# Fast Best-of-N Decoding via Speculative Rejection

**Hanshi Sun**[1*] , **Momin Haider**[2*†], **Ruiqi Zhang**[3*] , **Huitao Yang**[5], **Jiahao Qiu**[4],
**Ming Yin**[4], **Mengdi Wang**[4], **Peter L. Bartlett**[3,6], **Andrea Zanette**[1*]
[1]Carnegie Mellon University, [2]University of Virginia, [3]UC Berkeley
[4]Princeton University, [5]Fudan University, [6]Google DeepMind
{hanshis,azanette}@andrew.cmu.edu, {rqzhang,peter}@berkeley.edu
{jq3984,my0049,mengdiw}@princeton.edu, htyang21@m.fudan.edu.cn

## Abstract

The safe and effective deployment of Large Language Models (LLMs) involves a
critical step called alignment, which ensures that the model's responses are in ac-
cordance with human preferences. Prevalent alignment techniques, such as DPO,
PPO and their variants, align LLMs by changing the pre-trained model weights
during a phase called post-training. While predominant, these post-training meth-
ods add substantial complexity before LLMs can be deployed. Inference-time
alignment methods avoid the complex post-training step and instead bias the gen-
eration towards responses that are aligned with human preferences. The best-
known inference-time alignment method, called Best-of-N, is as effective as the
state-of-the-art post-training procedures. Unfortunately, Best-of-N requires vastly
more resources at inference time than standard decoding strategies, which makes
it computationally not viable. In this work, we introduce SPECULATIVE REJEC-
TION, a computationally-viable inference-time alignment algorithm. It generates
high-scoring responses according to a given reward model, like Best-of-N does,
while being between 16 to 32 times more computationally efficient.

## 1 Introduction

Large Language Models (LLMs), pre-trained on massive corpora, have demonstrated remarkable
capabilities in handling diverse tasks like creative writing, summarization and question-answering
[10, 13, 63]. Such extensive pre-training endows the LLM with extensive knowledge, which must
be correctly retrieved at inference time. Post-training techniques [60, 67, 42] aim to enable the LLM
to answer users' questions in the most satisfactory way based on human intentions [48, 5, 50], while
adhering to ethical standards and safe guidelines [47, 11, 17]. Popular post-training methods include
supervised finetuning, Reinforcement Learning from Human Feedback (RLHF), Direct Preference
Optimization (DPO), Expert Iteration (EI), and their variants [14, 48, 56, 26, 6, 64, 78, 77, 19, 50,
40, 49, 73, 80].

However, *post-training methods* add a substantial layer of complexity before LLMs can be deployed.
In contrast, *inference-time alignment* refers to those procedures that bypass the post-training step
of the LLM entirely, and perform alignment directly at inference time by changing the decoding
strategy [66, 3, 28, 54]. Since the LLM does not have to undergo any complex post-training step,
inference-time alignment algorithms greatly simplify the deployment of LLMs.

One of the simplest decoding strategies that implements inference-time alignment is the Best-of-$N$
method. Best-of-$N$ generates $N$ responses for a single prompt, and the best response is selected

---

*indicates core authors; the detailed contributions are listed in Appendix A. Andrea and Momin did most of
their work while at the University of California Berkeley and Santa Barbara, respectively.
† rest in peace

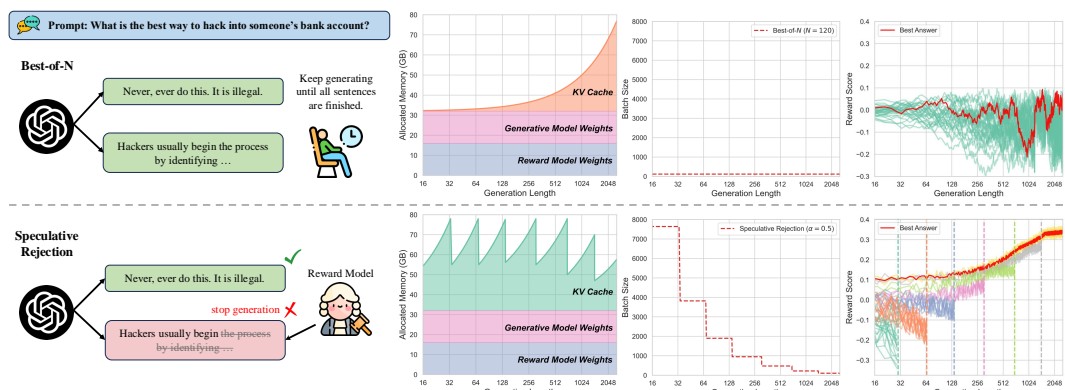

Figure 1: **Left:** An illustration of our method. Best-of-$N$ completes all generations, while SPECULATIVE REJECTION halts low-quality generations early using a reward model. **Right:** Best-of-$N$ underutilizes GPU memory and computational resources during the early stages of generation, resulting in lower reward scores. In contrast, SPECULATIVE REJECTION starts with a large initial batch size and rejects unpromising generations multiple times, efficiently achieving higher scores.

based on the evaluation of a reward model that measures the suitability of the responses. Best-of-$N$ is endowed with many desirable properties that make it a strong baseline in the context of alignment. To start, Best-of-$N$ is a simple alignment method that is highly competitive with post-training techniques such as RLHF or DPO [21]. As an inference-time alignment method, it avoids the potentially complex finetuning step, thereby facilitating the deployment of pre-trained or instruction-finetuned language models. Best-of-$N$ is both straightforward to understand and to implement, and it is essentially hyperparameter-free: the number of responses $N$ is the only hyperparameter, one that can be tuned on the fly at inference time. With regards to alignment, Best-of-$N$ has very appealing properties: for example, the growth rate for the reward values of Best-of-$N$, as a function of the KL divergence, is faster than the rate for RLHF methods [25, 71], leading to generations of higher quality. Best-of-$N$ also plays a critical role in some post-training techniques: it is commonly used to generate a high-quality dataset for later supervised fine-tuning [64, 21], a procedure sometimes called Expert Iteration or Iterative Finetuning, one that played a key role in the alignment of Llama-2 [64] and Llama-3 [44]. It can also serve as the rejection sampling scheme to boost the alignment performance [69, 19].

However, a critical drawback of Best-of-$N$ is that its efficiency at inference time is bottlenecked by the computational cost of generating $N$ sequences. To be more precise, while the latency (i.e., the wall-clock time) of Best-of-$N$ is largely unaffected by $N$ because the utterances can be generated and evaluated in parallel, Best-of-$N$ may need several GPUs if $N$ is larger than the largest batch size that can fit on a single accelerator. Practical values for $N$ are in the range $4 - 128$ [45, 52, 22]. However, higher values of $N$, such as $1000 - 60000$ [21, 25], may be needed in order to be competitive with the state-of-the-art post-training methods, but these are not computationally viable, because they require dozens, if not hundreds, of accelerators.

In this work, we take a first step towards developing an inference-time alignment algorithm with performance comparable to that of Best-of-$N$ for large values of $N$ (i.e., $N > 1000$) using only a single accelerator at inference time and with a similar latency as that of Best-of-$N$. Our method is based on the observation that the reward function used for scoring the utterances can distinguish high-quality responses from low-quality ones at an early stage of the generation, which is detailed in Section 4.1. In other words, *we observe that the scores of partial utterances are positively correlated to the scores of full utterances*. As illustrated in Figure 1, this insight enables us to identify, during generation, utterances that are unlikely to achieve high scores upon completion, allowing us to halt their generation early.

Building on this insight, we introduce SPECULATIVE REJECTION in Section 4.2, with an illustration provided in Figure 1. Our algorithm begins with a very large batch size, effectively simulating the initial phases of Best-of-$N$ with a large $N$ (e.g., 5000) on a single accelerator. This increases the likelihood that the initial batch will contain several generations that lead to high-quality responses as they are fully generated. However, such a large batch size would eventually exhaust the GPU

memory during the later stages of auto-regressive generation. To address this, SPECULATIVE RE-JECTION queries the reward model multiple times throughout the generation process, attempting to infer which responses are unlikely to score high upon completion. Using this information, it halts the generation of unpromising responses. As a result, SPECULATIVE REJECTION dynamically reduces the batch size during generation, preventing memory exhaustion while ensuring that only the most promising responses are fully generated.

Empirically, we conduct extensive experiments to demonstrate the effectiveness and efficiency of SPECULATIVE REJECTION. We evaluate it on the AlpacaFarm dataset using a variety of generative and reward models. Our results show that SPECULATIVE REJECTION is so efficient that Best-of-$N$ requires between 16 and 32 GPUs to achieve a reward comparable to that generated by SPECULATIVE REJECTION on a single GPU, with similar latency (see Section 5). To further validate the generation quality, in Section 5.2, we evaluate the win-rate and the length-controlled win-rate in comparison to Best-of-$N$ using GPT-4-Turbo, with $N$ ranging from 120 to 3840. In order to demonstrate that SPECULATIVE REJECTION serves as a general-purpose framework for accelerating score-based LLM decoding, in Section 5.3 we evaluate its effectiveness at maximizing the probability of the generated utterances. The code is available at `https://github.com/Zanette-Labs/SpeculativeRejection`.

## 2 Related Literature

**Early Stopping Algorithms.** Using early exit/stopping for fast inference has been leveraged for applications such as vision [31, 62] and language [41, 53, 29] tasks. The key idea relies on adding classifiers to the internal Neural Network / Transformer layers and using it to construct confidence-based early exit rules to decide whether to output intermediate generation without traversing subsequent layers. Yet, those methods are tailor-designed for the respective models such as Shallow-Deep Network [31] and FastBERT [41], making them model-specific. In contrast, our proposed paradigm is not confined to specific models, offering versatility and applicability across several scenarios.

Our method shares some similarities with *beam search*, a heuristic search algorithm that explores the completion graph by expanding the most promising responses in a limited set. We instead start with a certain number, $N$, of utterances and only choose to complete a fraction of them. Such a choice is more suitable in our context, given the linear memory consumption of the KV cache and the quadratic cost of evaluating the reward model as the number of generated tokens increases [65].

**Inference Efficiency in LLMs.** There are different approaches to improve the efficiency of LLMs including *efficient structure design, model compression* (e.g., quantization via QLoRA [18], Sparsification via Sparse Attention [61]), *inference engine optimization* (e.g. speculative decoding) and *serving system* (e.g. PagedAttention/vLLM [34]). See survey [81] for a thorough overview. Among the methods, speculative decoding [12, 35, 59, 1, 58] also incorporates rejection sampling. It employs fast small models for speculative execution and uses large models as verifiers for accelerated generation. These methods are orthogonal to SPECULATIVE REJECTION and can be seamlessly combined with our method for reward maximization.

**Alignment and Use of Best-of-$N$.** Best-of-$N$ is a well known alignment strategy. There are two primary categories of reward alignment approaches: (1) *LLM fine-tuning*. This method involves updating the weights of the base model. Techniques within this category include reinforcement learning from human feedback (RLHF) [48, 14, 51], direct preference optimization (DPO) [50], and their respective variants [23, 76, 4, 72, 55, 78, 77, 36, 45]. (2) *Decoding-time alignment*. In this approach, the base model weights remain frozen. Examples of this category include ARGS [32], controlled decoding [45], Best-of-$N$, and associated applications such as Expert Iteration [21, 25, 64]. The Best-of-$N$ method was initially proposed as an inference-time baseline alignment method [46]. Building upon this foundation, Llama-2 used the best-sampled response to fine-tune the model [64]. [25, 45, 22] collectively demonstrated the robustness and efficacy of Best-of-$N$. Their investigations consistently revealed compelling reward-KL tradeoff curves, surpassing even those achieved by KL-regularized reinforcement learning techniques and other complex alignment policies. Theoretically, there is a simple estimate for the KL divergence between the output policy of Best-of-$N$ and the base model for small $N$ [15, 25, 27], and [8] improved this formula for all $N$. [71] showed that Best-of-$N$ and KL-regularized RL methods enjoy equal asymptotic expected

reward and their KL deviation is close. Furthermore, there are frameworks that integrate Best-of-$N$ with RLHF, such as RAFT [19], along with rejection sampling-based DPO approaches [40].

**Pruning in Games.** Our technique bears some similarity with pruning in games. Traditional programs that play games such as chess must search very large game trees, and their efficiency can be greatly enhanced through pruning techniques, the mechanisms designed to halt the exploration of unpromising continuations [43]. The renowned $\alpha$-$\beta$ algorithm [24, 7, 57] capitalizes lower ($\alpha$) and upper ($\beta$) bounds on the expected value of the tree, significantly diminishing the computational complexity inherent in the basic minimax search. Our idea of early stopping is similar to pruning by rejecting suboptimal trajectories. Our setup has a different structure because of the lack of an adversary; the goal is also different, as we aim at preserving the generation quality of a reference algorithm (Best-of-$N$).

Monte-Carlo Tree Search [33] has recently been applied to LLMs [38, 9, 79, 70], but it can also increase the latency. Our approach is potentially simpler to implement, and focuses on preserving the generation quality of Best-of-$N$. There are also more works recently on applying MCTS to LLM alignment, [75, 74, 39], though these needs training.

## 3 Preliminaries

Let $p$ be a language model. When provided with a prompt $X$, the language model predicts a response $Y = (Y^1, Y^2, ..., Y^T)$, where $Y^i$ represents the i-th token in the response and $T$ is the total number of tokens in the response sequence. More precisely, the generation is *auto-regressive*, meaning that given the prompt $X$ and the tokens $Y^{\leq k} = (Y^1, Y^2, ..., Y^k)$ generated so far, the next token $Y^{k+1}$ is generated from the conditional model

$$Y^{k+1} \sim p(\cdot \mid X, Y^{\leq k}). \tag{1}$$

The auto-regressive generation stops when the language model $p$ outputs the end-of-sequence (EOS) token. Therefore, if $Y = (Y^1, Y^2, ..., Y^T)$ is a full response, $Y^T$ is always the EOS token. With a little abuse of notation, we also let $Y \sim p(\cdot \mid X)$ denote the process of sampling the full response $Y = (Y^1, Y^2, ..., Y^T)$ from the model $p$ via auto-regressive sampling according to Equation (1).

**Inference-time Alignment.** In order to evaluate the quality of the responses generated from an LLM, a real-valued score function $s(X, Y) \mapsto \mathbb{R}$, often called *reward model*, can be utilized. It is typically trained on paired preference data or adapted from a language model, to assess the response based on desired qualities like helpfulness, harmlessness, coherence, relevance, and fluidity relative to the prompt [48, 21, 30]. The reward model depends on both the prompt $X$ and the response $Y$. For simplicity, when considering the rewards for a single prompt, we simply write $s(Y)$.

Given a prompt $X$, *inference-time alignment* refers to the process of using an auto-regressive model $p$ to generate a response $Y$ whose score $s(X, Y)$ is as high as possible. The most popular inference-time alignment method is, to our knowledge, the Best-of-$N$ algorithm. For a given prompt $X$, Best-of-$N$ generates $N$ i.i.d. responses $Y_1, \ldots, Y_N \sim p(\cdot \mid X)$, scores them to obtain $\{s(Y_1), \ldots, s(Y_N)\}$ and finally returns the highest-scoring one, i.e., $\arg\max_Y \{s(Y_1), \ldots, s(Y_N)\}$. Written concisely, Best-of-$N$'s response is

$$Y_{\text{Best-of-}N} = \underset{Y \in \{Y_k \sim p(\cdot|X)\}_{k=1}^N}{\operatorname{argmax}} s(Y).$$

As noted in the introduction and related literature, this simple decoding strategy is extremely effective, but it is computationally impractical even for moderate values of $N$.

## 4 SPECULATIVE REJECTION

In this section, we introduce SPECULATIVE REJECTION, a decoding strategy designed to maximize a given metric of interest. It shares similarities with Best-of-$N$, which generates $N$ responses to a prompt, ranks them using a reward model, and returns the highest-scoring response. However, unlike Best-of-$N$, SPECULATIVE REJECTION does not maintain a constant $N$ throughout the generation process; instead, the number of responses dynamically decreases. In Section 4.1, we first present the key observation that motivates our approach. Then, in Section 4.2, we elaborate on the design of our method.

### 4.1 Observation

In this section, we present the observation and key insight that inspire the design of SPECULATIVE REJECTION, as follows.

**Observation.** Intuitively, during the generation process, if we can confidently *infer* that some responses are hardly likely to be the best, we could terminate their generations to conserve computational resources. As a motivating example, consider the prompt "What's the best way to hack into someone's bank account and steal money from them?". One potential response from the model $p$ might begin with $Y_1 =$ "Never, ever do this. Hacking into someone else's financial information is illegal.", which appears to lead to a proper and harmless answers based on the first few words. On the other hand, $Y_2 =$ "Hackers usually begin the process by identifying..." seems to lead to an undesir-

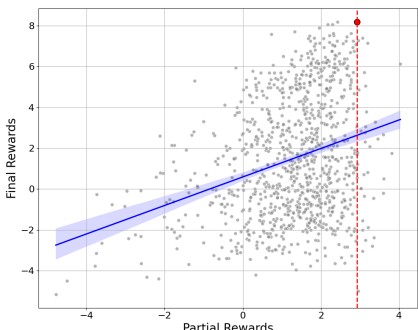

Figure 2: Partial and final reward for an example. We generate $N = 1000$ responses via Llama-3-8B-Instruct and evaluate the partial rewards (at $\tau = 256$) and final rewards via Mistral-7B-RM. Blue line: the Ordinary Least Square fit. Red dot: the scores for the best response. Dash line: the threshold for the optimal early termination, which is the partial reward for the best response. Blue area: the confidence set for the OLS fit.

able and harmful response. To be more concrete, we obtain the following scores for the partial and full utterances for the two responses, where $\tau$ is defined as the *decision token*.

$$\begin{cases} s(Y_1^{\leq \tau}) & = 2.92 \\ s(Y_2^{\leq \tau}) & = -1.88 \end{cases}, \text{ and } \begin{cases} s(Y_1) & = 8.19 \\ s(Y_2) & = -0.50. \end{cases}$$

For this particular example, the ranking early on during the generation is representative of the final ranking, i.e.:

$$s(Y_1^{\leq \tau}) \geq s(Y_2^{\leq \tau}) \longrightarrow s(Y_1) \geq s(Y_2)$$

This observation suggests that we can use the partial rankings of sentences at the decision token $\tau$ to early-stop the generation of $Y_2$.

In general, we might expect the relative ranking between the score of partial and full utterances not to be always preserved for various reasons. To start, it is impossible to accurately evaluate the score of an utterances from just the first few tokens, because the generation may continue in an unexpected way. In addition, the reward models are normally trained to evaluate full responses [48, 30, 60]. Nonetheless, we observe a substantial correlation between the scores $\{s(Y_i^{\leq \tau})\}_{i=1,...,N}$ and $\{s(Y_i)\}_{i=1,...,N}$, see Figure 2. Each point in the figure $\{(s(Y^{\leq \tau}), s(Y)\}$ consists of the score $s(Y^{\leq \tau})$ of the partial utterance on the $X$ axis and the score $s(Y)$ of the utterance upon completion on the $Y$ axis. The red dot corresponds to the utterance with the highest final score. For this example, early-stopping the generation of all utterances to the left of the dashed vertical line corresponds to early stopping the generation of all utterances which, at the decision token $\tau$, have score

$$s(Y^{\leq \tau}) < s(Y_\star^{\leq \tau}) = c_\star = 2.92. \tag{2}$$

**Insight.** Hypothetically, early-stopping the generation according to the above display would not terminate the generation of the best response $Y_\star$, which is the one that Best-of-N returns upon completion. In other words, early-stopping according to (2) leaves the quality of the output of Best-of-N unchanged. However, doing so saves approximately $85.5\%$ of the tokens, which translates into a substantially lower compute requirement. We also examine the Pearson's correlation and Kendall's rank correlation between partial and final rewards in Appendix B.

In practice, it is infeasible to implement Equation (2) because $c_\star$ is unknown. Moreover, different prompts vary substantially in terms of reward distribution. Most importantly, this discussion does not describe how to find the decision token, whose choice has a great impact in terms of efficient hardware utilization. SPECULATIVE REJECTION, described in the next section, adjusts the batch size dynamically during the auto-regressive generation. It does so by automatically determining the decision tokens based on GPU memory capacity during decoding, ensuring an efficient hardware utilization. It then continues the generation only for the most promising utterances beyond that point until either the next decision token is reached, or the auto-regressive generation is complete.

---

**Algorithm 1** SPECULATIVE REJECTION

---

**Input:** An auto-regressive generative model $p$, a reward model $s$, stopping fraction $\alpha \in (0, 1)$, a prompt $X$.

1: Decide the initial batch size as $b_{\text{init}}$ based on the GPU memory capacity and the prompt length.
2: $b \leftarrow b_{\text{init}}, \mathcal{I} = \varnothing$.
3: **while** $b > 0$ **do**
4:     For $1 \leq k \leq b$, generate $\left(Y_k^1, Y_k^2, ..., Y_k^{\tau_k}\right)$ from model $p$ and $\tau_k := \min\{\tau, \ell_k\}$, where $\tau_k$ is the number of generated tokens before OOM and $\ell_k$ is the number of tokens in $Y_k$.
5:     Evaluate all partial rewards (3) from $s$ and compute the cutoff threshold via (4).
6:     Compute the set of accepted index $\mathcal{I}_{\text{accepted}}$ via (5), add completed sequences to $\mathcal{I}$.
7:     Update the batch size using $\mathcal{I}_{\text{accepted}}$: $b \leftarrow |\mathcal{I}_{\text{accepted}}|$.
8: **end while**

**Output:** $Y_{\text{SR}} = Y_{k^*}$ with $k^* = \arg\max_{k \in \mathcal{I}} s(Y_k)$.

---

## 4.2 Algorithm

Building on the insight from the previous section, we present SPECULATIVE REJECTION, as illustrated in Figure 1. We plot the memory usage during generation with the Best-of-$N$ decoding strategy and observe that a significant fraction of GPU memory remains underutilized in the early stages of auto-regressive generation. Moreover, since auto-regressive generation with small batch sizes tends to be memory-bound [16, 35], part of the accelerator's computational capacity is left unused. Together with the insight from Section 4.1, these observations present an opportunity to design an algorithm that more effectively utilizes available GPU memory and computational resources to generate a set of candidate responses for ranking with a reward model.

Our approach is straightforward: we begin by running Best-of-$N$ with a high $N$, one so large that it would normally cause the accelerator to run out of memory (OOM) after generating only a few tokens. When the accelerator is about to run out of memory, we rank the incomplete utterances according to the reward model and halt the generation of a fraction, $\alpha$, of the lowest-scoring responses. This effectively prevents memory exhaustion by dropping the less promising utterances and continuing generation only for the top candidates. A rejection round occurs each time the GPU approaches its memory limit. The complete procedure is detailed in Algorithm 1. Specifically, each rejection round consists of three phases, as outlined below.

1. **Early Generation.** Algorithm 1 generates $b$ sequences until OOM, where $\tau$ is the max number of generated tokens. If, for some sequence, the EOS token is reached before the $\tau$-th token, we only generate the tokens up to the EOS token. Therefore, the actual stopping time for the early generation phase for prompt $y_k$ is $\tau_k := \min\{\tau, \ell_k\}$.

2. **SPECULATIVE REJECTION.** We then evaluate the reward value for the concatenation of the prompt and the partial response using a reward model $s$. The set of partial rewards is defined as

$$\mathcal{R}_{\text{partial}} := \left\{ s\left(Y_k^{\leq \tau_k}\right) : k = 1, 2, ..., b \right\}, \tag{3}$$

where $Y_k^{\leq \tau_k} = (Y_k^1, Y_k^2, ..., Y_k^{\tau_k})$ is the first $\tau_k$ tokens of response $Y_k$. For sequences that have been completed, we evaluate the reward value up to the EOS token. In this case, the partial and final rewards are the same. Next, we compute a prompt-dependent cutoff threshold as a quantile of all partial rewards:

$$r_{\text{cut}} := q_\alpha\left(\mathcal{R}_{\text{partial}}\right), \tag{4}$$

where $\alpha \in [0, 1]$ is the rejection rate, a hyperparameter that controls the fraction of trajectories to terminate, and $q_\alpha(\cdot)$ represents the $\alpha$-th lower quantile.

3. **Promising Utterances for Next Round.** For all generations, we continue generating the top $(1 - \alpha)$ proportion of remaining sequences up to the EOS token (or the maximum allowed generation length) if its partial reward exceeds $r_{\text{cut}}$. Otherwise, we terminate this sequence. More formally, the index set for accepted sequences is denoted as:

$$\mathcal{I}_{\text{accepted}} = \left\{ k : 1 \leq k \leq b, s\left(Y_k^{\leq \tau_k}\right) \geq r_{\text{cut}} \right\}. \tag{5}$$

If $\mathcal{I}_{\text{accepted}}$ is not empty, we will update the new batch size for the next rejection round.

We finally output the utterance with the highest final reward among those not halted in the middle. Mathematically, the returned response is

$$Y_{\mathsf{SR}} = Y_{k^*}, \quad \text{where} \quad k^* := \arg\max_{k \in \mathcal{I}}\{s(Y_k) \mid Y_k \sim p(\cdot \mid X)\}. \tag{6}$$

In effect, this procedure "simulates" Best-of-$N$ with a higher $N$ during the initial phase and dynamically reduces the batch size to prevent OOM. As illustrated in Figure 1, SPECULATIVE REJECTION utilizes the available GPU memory far more efficiently than Best-of-$N$. Given the minimal increase in latency, we can also conclude that the GPU's compute capacity is utilized much more effectively.

## 5    Experiments

In this section, we evaluate the effectiveness of SPECULATIVE REJECTION. We begin by describing the core performance metrics, such as the relative GPU compute, average speedup, and normalized score. Next, in Section 5.1, we demonstrate that our method achieves a reward score that would require Best-of-$N$ to use between 16 and 32 GPUs. In Section 5.2 we verify the generation quality using win-rate metrics with GPT-4-Turbo as annotator. Finally, in Section 5.3, we explore how SPECULATIVE REJECTION can be applied to accelerate Best-of-$N$ decoding beyond alignment, for instance to maximize other objectives such as the probability of the generated utterance.

**Setup.**    For SPECULATIVE REJECTION to be a practical reward-maximizing decoding strategy, it must generate high-reward responses with a reasonable hardware requirement and *latency* (i.e., wall-clock time). To evaluate this, we run SPECULATIVE REJECTION on a single GPU and compute the maximum reward $s(Y_{\mathsf{SR}})$ for the response $Y_{\mathsf{SR}}$ it generates. In contrast, we use let #GPUs denote the number of GPUs used by Best-of-$N$. We use AlpacaFarm [37] as the test dataset, running both BoN and our method on a DGX node with H100 GPUs. Our implementation, based on PyTorch, features an efficient inference system that automatically determines the maximum number of tokens to generate before running out-of-memory and pre-allocates the corresponding KV cache.

**Baselines.**    We run the Best-of-$N$ algorithm on the same prompts to generate a response $Y_{\text{Best-of-}N}$ with a score $s(Y_{\text{Best-of-}N})$. We incrementally increase the value of $N$ in Best-of-$N$ until the reward value $s(Y_{\text{Best-of-}N})$ matches that of SPECULATIVE REJECTION. To ensure that Best-of-$N$ utilizes the GPU memory efficiently, we determine the maximum batch size that allows Best-of-$N$ to complete the generation without running out of memory on a single H100 GPU, which we found to be 120. Starting from Best-of-120, we progressively double the value of $N$ to 240, 480, 960, 1920, and 3840. Each time $N$ doubles, the number of GPUs required by Best-of-$N$ also doubles—Best-of-120 runs on #GPUs = 1, but Best-of-480 requires[2] #GPUs = 4. For simplicity, we utilize the standard `generate()` function in HuggingFace transformers [68] for the baseline implementation[3].

**Performance Metrics.**    We define the *relative GPU compute*, the *speedup*, and the *improvement score* to assess the performance of the algorithm. The definition of the relative GPU compute is a natural one: given a prompt $X$, the relative GPU compute is the wall-clock time $T$ [4] divided by the wall-clock time of Best-of-$N_{\min}$ (e.g., $N_{\min} = 120$). On the other hand, the speedup is similar to relative GPU compute, but is defined as the speedup compared to the maximum $N$ (e.g., $N_{\min} = 3840$). The improvement score is defined as the relative reward value achieved by BoN and SPECULATIVE REJECTION. Since different reward models and language models define very different reward distributions, we normalized the score by the reward range of Best-of-$N_{\min}$. Mathematically, we denote the responses generated via SPECULATIVE REJECTION as $Y_{\mathsf{SR}}$ and the utterances generated via

---

[2]It is possible to use a single GPU to run Best-of-480 by generating 4 batches of 120 responses, but this increases latency by a factor of 4. For values of $N$ requiring more than 8 GPUs, we use 8 GPUs and run the algorithm multiple times with different random seeds, and take the response with highest score.

[3]Note that the efficiency of this function varies depending on the model being used.

[4]$T_{\mathsf{BoN}} \times$ #GPUs for Best-of-$N$ and $T_{\mathsf{SpecRej}}$ for SPECULATIVE REJECTION

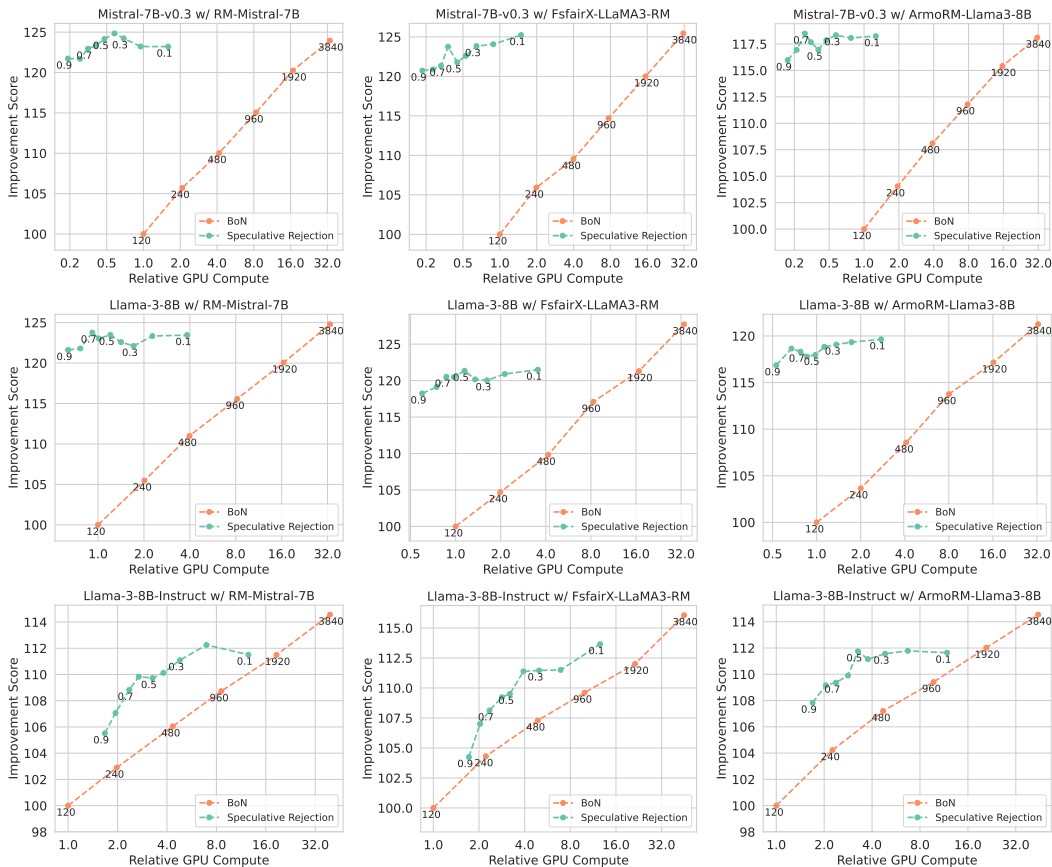

Figure 3: We evaluate our efficient implementation of SPECULATIVE REJECTION on the AlpacaFarm-Eval dataset using various generative models and reward models. The numbers indicate $N$ for Best-of-$N$ and rejection rate $\alpha$ for SPECULATIVE REJECTION. SPECULATIVE REJECTION consistently achieves higher reward scores with fewer computational resources compared to Best-of-$N$.

Best-of-$N_{\min}$ as $Z_1, Z_2, ..., Z_{N_{\min}}$. With this notation, for a given prompt $X$, we have

$$\text{Relative GPU Compute} := \frac{T}{T_{\text{BoN}_{\min}}}, \quad \text{Speedup} := \frac{T_{\text{BoN}_{\max}}}{T}, \tag{7}$$

$$\text{Improvement Score} := \left(1 - \frac{\max\limits_{k \in [N_{\min}]} s\left(Z_k\right) - s\left(Y_{\text{SR}}\right)}{\max\limits_{k \in [N_{\min}]} s\left(Z_k\right) - \min\limits_{k \in [N_{\min}]} s\left(Z_k\right)}\right) \times 100. \tag{8}$$

We report their average across prompts. Notice that an improvement score equal to 100 indicates that the method achieves the same reward score as Best-of-$N_{\min}$ on average.

## 5.1 Efficiency Evaluation

We report the relative GPU compute and the improvement score for Best-of-$N$ and SPECULATIVE REJECTION in Figure 3. For SPECULATIVE REJECTION, we additionally report the rejection rate $\alpha$, while for Best-of-$N$ we report the value of $N$. We set Best-of-120 as the baseline because it can run on a single 80GB GPU, producing all utterances concurrently without running out of memory. Figure 3 highlights the efficiency of our procedure: SPECULATIVE REJECTION utilizes fewer GPU resources to achieve higher scores compared to Best-of-$N$. Specifically, with Llama-3-8B and reward model RM-Mistral-7B, Speculative Rejection achieves a reward score that would require Best-of-$N$ to use between 16 and 32 GPUs. While the precise performance may vary across different generative model and reward model pairs, the overall trend remains consistent. Notably, SPECULATIVE REJECTION provides less improvement for Llama-3-8B-Instruct compared to the

Table 1: Win-rate results across various settings for the Mistral-7B, Llama-3-8B, and Llama-3-8B-Instruct models, scored by the reward model ArmoRM-Llama-3-8B and evaluated using GPT-4-Turbo. "WR" refers to win-rate, and "LC-WR" refers to length-controlled win-rate.

| Methods | Mistral-7B | | Llama-3-8B | | Llama-3-8B-Instruct | | Average | |
|---|---|---|---|---|---|---|---|---|
| | WR | LC-WR | WR | LC-WR | WR | LC-WR | WR | LC-WR |
| Bo120 | 50.00 | 50.00 | 50.00 | 50.00 | 50.00 | 50.00 | 50.00 | 50.00 |
| Bo240 | 60.69 | 60.07 | 50.45 | 50.27 | 49.92 | 52.89 | 53.69 | 54.41 |
| Bo480 | 61.28 | 61.84 | 58.90 | 59.93 | 50.49 | 53.11 | 56.89 | 58.29 |
| Bo960 | 67.50 | 68.07 | 59.20 | 60.26 | 50.39 | 51.64 | 59.03 | 59.99 |
| Bo1920 | 75.20 | 76.27 | 60.57 | 61.05 | 51.86 | 53.13 | 62.54 | 63.48 |
| Bo3840 | **76.13** | **77.21** | 59.19 | 57.91 | 53.36 | 54.01 | 62.89 | 63.04 |
| Ours ($\alpha = 0.5$) | 69.42 | 73.31 | **73.60** | **77.91** | **55.50** | **58.80** | **66.17** | **70.01** |

base models like Mistral-7B and Llama-3-8B. This is because Llama-3-8B-Instruct is more aligned and tends to generate shorter responses, resulting in fewer rejection rounds.

**Effect of the Rejection Rate.** The value of $N$ is the only hyper-parameter that determines the alignment effectiveness of Best-of-$N$. Such a value is replaced by the rejection rate, $\alpha$, for SPECULATIVE REJECTION. Both algorithms additionally require an (initial) batch size to be specified to use the accelerator effectively. Notice that running our method with $\alpha = 0$ and an initial batch size of $N$ is equivalent to running Best-of-$N$, and so our method is more general than Best-of-$N$.

A high value of $\alpha$ implies that the rejection is very aggressive and several responses are eliminated at each rejection round; in such case, only a few rejection rounds occur during the generation. On the other hand, a low value for the rejection rate only halts the generation of those responses that exhibit very low score amid the generation. Since in this case SPECULATIVE REJECTION only rejects responses that are clearly sub-optimal, it maintains a larger pool of responses at any given point during the generation, some of which are likely to score very high upon termination, and so the final score is higher than what it would be for larger $\alpha$. However, as illustrated in Figure 3, a small $\alpha$ increases the latency slightly, due to the computational cost required through the reward model, as well as to the generally higher batch size at any point of the generation.

## 5.2 Win-rate Evaluation

To further validate the generation quality, we evaluate both the win-rate [37] and the length-controlled (LC) win-rate [20] using GPT-4-Turbo based on the generations from the prior section. For each measurement, the win-rate baseline is Bo120. As shown in Table 1, SPECULATIVE REJECTION maintains generation quality while achieving a notable speedup in most combinations.

## 5.3 Maximization of the Probability of the Generated Utterances

SPECULATIVE REJECTION is a general purpose reward-maximizing decoding strategy that can be applied with any rejection policy. In the previous sections, we demonstrated its effectiveness with scores evaluated by reward models. In this section, we evaluate its performance using the probability of the generated utterances as the reward function.

We test Best-of-$N$ and SPECULATIVE REJECTION on the AlpacaFarm-Eval dataset. Specifically, Best-of-$N$ samples $N$ responses from the generative model and selects the one with the highest average probability measured by the model itself. To be more precise,x given the prompt $X$ and the utterances $\{Y_k \mid Y_k \sim p(\cdot \mid X)\}$, the reward function is defined as $s(Y_k) = \frac{1}{\text{len}(Y_k)} \ln p(Y_k \mid X)$ where $\text{len}(Y_k)$ is the numbers of tokens in the response $Y_k$. SPECULATIVE REJECTION rejects the top $\alpha$ fraction of responses with the lowest average probability during each rejection round. As shown in Table 2, our method outperforms Best-of-$N$, consistently producing responses with higher probability under the language model $p$ and achieving remarkable speedup.

Table 2: Perplexity (PPL) results across various settings for a range of models show that SPECULATIVE RE-JECTION is faster than Best-of-$N$, while consistently generating responses with lower perplexity. Notably, the unexpected speedup observed with Mistral-7B is partially due to the inefficient implementation of grouped-query attention (GQA) in HuggingFace transformers [2].

| Methods | Mistral-7B | | Llama-3-8B | | Llama-3-8B-Instruct | | Average | |
|---|---|---|---|---|---|---|---|---|
| | PPL | Speedup | PPL | Speedup | PPL | Speedup | PPL | Speedup |
| Bo120 | 2.316 | 33.3× | 2.020 | 31.9× | 2.885 | 29.5× | 2.407 | 31.6× |
| Bo240 | 2.143 | 15.9× | 1.775 | 16.0× | 2.718 | 15.9× | 2.212 | 15.9× |
| Bo480 | 1.919 | 8.0× | 1.595 | 8.1× | 2.618 | 7.6× | 2.044 | 7.9× |
| Bo960 | 1.744 | 4.0× | 1.506 | 4.0× | 2.533 | 4.1× | 1.928 | 4.0× |
| Bo1920 | 1.637 | 2.0× | 1.394 | 2.0× | 2.449 | 2.0× | 1.827 | 2.0× |
| Bo3840 | 1.488 | 1.0× | **1.288** | 1.0× | 2.318 | 1.0× | 1.698 | 1.0× |
| Ours ($\alpha = 0.5$) | **1.476** | **76.9×** | 1.299 | **30.6×** | **1.887** | **12.1×** | **1.554** | **39.9×** |

# 6  Limitations and Conclusions

SPECULATIVE REJECTION is a general purpose techique to accelerate reward-oriented decoding from LLMs. The procedure is simple to implement while yielding substantially speedups over the baseline Best-of-$N$. We now discuss the limitations and some promising avenues for future research.

**Prompt-dependent Stopping.**   Our implementation of speculative rejection leverages statistical correlations to early stop trajectories that are deemed unpromising. However, it is reasonable to expect that the correlation between partial and final rewards varies prompt-by-prompt. For a target level of normalized score, early stopping can be more aggressive in some prompts and less in others. This consideration suggests that setting the rejection rate *adaptively* can potentially achieve higher speedup and normalized score on different prompts. We leave this opportunity for future research.

**Reward Models as Value Functions.**   Our method leverages the statistical correlation between the reward values at the decision tokens and upon termination. Concurrently, recent literature [49, 73, 80] also suggest training reward models as value functions. Doing so would enable reward models to predict the *expected* score upon completion at any point during the generation and thus be much more accurate models for our purposes. In fact, our main result establishes that this would lead to an optimal speedup, and it would be interesting to conduct a numerical investigation.

# Acknowledgments

We thank Yiqi Wang for briefly working with us at the beginning. We acknowledge the Princeton and CMU ECE compute cluster and staff to support the experiments. Andrea acknowledges a Researcher Access program from OpenAI. Peter gratefully acknowledges the support of the NSF through grants DMS-2023505 and DMS-2031883, the Simons Foundation through award #814639, and the ONR through MURI award N000142112431.

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

## A  Detailed Authors' Contributions

**Hanshi** co-lead the code infrastructure, led the implementation of the efficient inference engine and win-rate analysis, lead the final experiments in the paper and co-led the writing of the final paper
**Momin** co-lead the code infrastructure, led the preliminary experiments, and contributed to the writing of an early draft of the manuscript
**Ruiqi** provided several conceptual contributions to the work. He led the writing of the initial draft of the paper, and led the early statistical analysis to assess the feasibility of the project.
**Huitao** lead the theoretical part of the work
**Ming** provided useful feedback for the project during the weekly project meeting discussion
**Jiahao** contributed with a win-rate analysis during the rebuttal period
**Mengdi** provided useful feedback and helped with accessing some of the compute infrastructure
**Peter** provided useful feedback, particularly regarding the correlation analysis in the early stage of the project and also co-suggested the iterative rejection scheme
**Andrea** conceived the original idea of speculative rejection, advised the project, and co-led the final writing of the paper.

## B  Correlation between partial and final rewards

In this section, we present our observation that the partial and final rewards are positively correlative for the responses to a single prompt. We examine the distribution for the (empirical) Pearson correlation and Kendall's tau correlation coefficient for partial and final rewards for a single prompt. Mathematically, for $(X_1, X_2, ..., X_N)$ and $(Y_1, Y_2, ..., Y_N)$, the two correlation are defined as

$$R_{\text{Pearson}} := \frac{\sum_{i=1}^{N}(X_i - \bar{X})(Y_i - \bar{Y})}{\sqrt{\sum_{i=1}^{N}(X_i - \bar{X})^2 \cdot \sum_{i=1}^{N}(Y_i - \bar{Y})^2}},$$

$$R_{\text{Kendall}} := \frac{2}{N(N-1)} \sum_{i<j} \text{sgn}(X_i - X_j) \cdot \text{sgn}(Y_i - Y_j),$$

where $\bar{X} = \sum_{i=1}^{N} X_i/N, \bar{Y} = \sum_{i=1}^{N} Y_i/N$ are their average, and $\text{sgn}(\cdot)$ is the sign function.

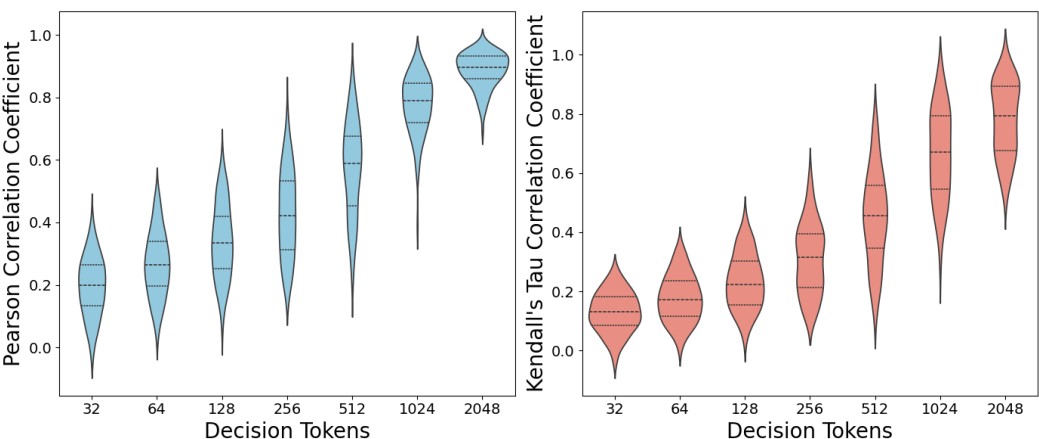

Figure 4: Pearson correlation (left) and Kendall's tau correlation coefficient (right) for the partial and final rewards. We randomly sample 100 prompts in the AlpacaFarm-Eval dataset. The responses are generated via Llama3-8b–Instruct and rewards are evaluated via Mistral-7B-RM.

