# OpenReview forum: "Fast Best-of-N Decoding via Speculative Rejection"
_NeurIPS.cc/2024/Conference — NeurIPS 2024 poster_

### Official Review · Reviewer_gUaJ · 2024-07-09

**Soundness:** 2
**Presentation:** 2
**Contribution:** 2
**Rating:** 5
**Confidence:** 3

**Summary:**

Best-of-N decoding involves generating multiple responses and then selecting the highest scoring one according to a given metric of interest. Based on the observation that the reward function used for scoring the utterances can distinguish high-quality hypotheses from low-quality ones at an early stage of the generation, this paper focuses on accelerating this procedure by stopping the generation of “unpromising utterances”, i.e., those that are unlikely to be returned at the end.

**Strengths:**

I summarize below the main strengths:
- The motivation is very clear and improving the efficiency of LLMs at inference time is an important problem.
- The empirical results seem to be promising.

**Weaknesses:**

Please see my questions and comments below. In my opinion, some parts of the submission could be significantly improved, including the overall organization of the paper, the related work section, and the discussion about situations in which the proposed method should or should not work. Also, I have some concerns about the experimental parts, which are detailed below.

**Questions:**

Comments and questions:

- L36-38: I don’t think that saying that Best-of-N is “essentially” hyperparameter-free makes sense here. As you point out, $N$ is a hyperparameter. Also, the hypotheses can be sampled in various ways (e.g., sampling with different temperatures, nucleus samplings, etc).
- There is a large body of work on decoding strategies for LLMs that rely on sampling multiple hypotheses and selecting the best one, including voting procedures [1], minimum bayes risk decoding [2], and other types of strategies [3, 4]. I believe they should at least be mentioned in the related work section.
- As briefly mentioned in L161-169, predicting scores for unfinished sentences may be misleading. Even though Figure 2 does not look bad, this is just for one example, right? Unless I’m missing something, this does not seem to be enough evidence to support your claims. Since this is key to your proposal, it would be beneficial to expand the discussion about this topic as well as the empirical evidence.
- L270-271: “we present experiments on the AlpacaFarm-Eval dataset, where we sample 100 prompts at random”. What do you mean by sampling 100 prompts? Is this a common practice?

Minor comments:
- Not all figures are referred to (see, e.g., Figures 5-14).
- Sometimes you refer to equations as “eq. (X)”, sometimes just as “(X)”. The same thing happens with “appendix X” and “Appendix X”, or “Figure X” and “fig. X”. Please try to be consistent throughout the paper.
- L306: Fix citation.

[1] Self-Consistency Improves Chain of Thought Reasoning in Language Models (Wang et al., ICLR 2023)

[2] It’s MBR All the Way Down: Modern Generation Techniques Through the Lens of Minimum Bayes Risk (Bertsch et al., BigPicture 2023)

[3] An Empirical Study of Translation Hypothesis Ensembling with Large Language Models (Farinhas et al., EMNLP 2023)

[4] Quality-Aware Decoding for Neural Machine Translation (Fernandes et al., NAACL 2022)

**Limitations:**

Yes.

---

> ### Author Rebuttal · Authors · 2024-08-07
>
> # Response for the reviewer gUaJ:
> We appreciate the reviewer's detailed comments and we have read them carefully. Here are our detailed responses.
>
> ---"L36-38: I don’t think that saying that Best-of-N is “essentially” hyperparameter-free makes sense here. As you point out,  is a hyperparameter. Also, the hypotheses can be sampled in various ways (e.g., sampling with different temperatures, nucleus samplings, etc)."---
>
> **Response:** This is a valid point. We agree that Best-of-N can rely on several hyperparameters, for example, temperature, nucleus sampling value (top-p), and top-k, among others. But it is also worth mentioning that sampling from language-models that went through a post-training phase also requires selecting the decoding strategy and its hyper-parameters (e.g., temperature) at decoding time. On top of these, Best-of-N only requires choosing N as additional parameter.
>
> Regarding the N hyperparameter, we examine its effect on our old and new experiments. When N is relatively small (~100-128), large rejection rates around 90% and higher have the potential to sacrifice significant performance. However, rejection rates as high as 80% still provide strong performance and improve efficiency at around 2-6x speedup, depending on the maximum length.
>
>
> ---"There is a large body of work on decoding strategies for LLMs that rely on sampling multiple hypotheses and selecting the best one, including voting procedures [1], minimum bayes risk decoding [2], and other types of strategies [3, 4]. I believe they should at least be mentioned in the related work section."---
>
> **Response:** We agree, the body of work on decoding strategies for LLMs is very large and only growing larger. Although our related work already contains several categories, we will also include references such as the ones above to highlight the variety of available decoding strategies and how they connect to our algorithm.
>
> ---"As briefly mentioned in L161-169, predicting scores for unfinished sentences may be misleading. Even though Figure 2 does not look bad, this is just for one example, right? Unless I’m missing something, this does not seem to be enough evidence to support your claims. Since this is key to your proposal, it would be beneficial to expand the discussion about this topic as well as the empirical evidence."---
>
> **Response:** We also include Figure 5 in Appendix D, which plots the correlation between partial reward and final reward across several decision tokens and for all prompts we tested. As shown,  there is positive correlation across prompts which allows our algorithm to work effectively.
>
> ---"L270-271: “we present experiments on the AlpacaFarm-Eval dataset, where we sample 100 prompts at random”. What do you mean by sampling 100 prompts? Is this a common practice?"---
>
> **Response:** Of the 805 prompts in the Alpaca Farm Eval dataset, we randomly sample (without cherry-picking) 100 of these prompts and perform all of our experiments on those 100 prompts, to save computation to run the baseline given the limited computational resources available to us.
>
> Also, since the win rate by GPT4 is expensive and OpenAI limits access to GPT4 on accounts each month, downsampling prompts from a large dataset is seen in other works [1].
>
> [1]Jang, Joel, et al. "Personalized soups: Personalized large language model alignment via post-hoc parameter merging." arXiv preprint arXiv:2310.11564 (2023).
>
> Regarding the minor comments, we will also improve the consistency when referring to equations, figures, etc. and fix any citations we need to. However, we should note that Figures 5-14 are part of the Appendix, and so are indirectly referenced in the main body via the Appendices.

---

> > ### Comment · Reviewer_gUaJ · 2024-08-09
> >
> > I've read the other reviews and the rebuttal. Thank you for answering my questions, I've updated my initial review accordingly.

---

### Official Review · Reviewer_qQm8 · 2024-07-10

**Soundness:** 4
**Presentation:** 4
**Contribution:** 3
**Rating:** 7
**Confidence:** 4

**Summary:**

Best-of-N is a decoding-time alignment algorithm that effectively aligns the output of the system at the cost of high inference time. The paper seeks to reduce the computation time by pruning unpromising sequences at early stage using the reward model to estimate the reward on the partial utterance. They empirically show that the estimate of the reward model on a partial sentence is correlated with the estimate of a complete sentence, using AlpacaFarm-Eval dataset. The proposed method achieves 2-8 times speedup at a marginal drop of the output quality.

**Strengths:**

- Proposed method is simple and effective.
- Experiments are clear and comprehensive, except for the variety of the tasks.
- Showing that partial rewards have good correlation with the final rewards (Figure 2) is a significant contribution to the community.

**Weaknesses:**

I don’t see any critical weaknesses in the study. Several minor points are listed below.

- Experiments are conducted only on AlpacaFarm-Eval dataset. It would be ideal to have an evaluation of the proposed method in other datasets (e.g., TL;DR, hh-rlhf) as the effectiveness of the method may be affected by the structure of the task. For example, I speculate the the reward of the partial text has less correlation with the complete text on tasks like text summarization. It may also be influenced by the language. For example, Korean and Japanese have more flexible word order than English and word order does not determine grammatical function. Thus, the same length of the first few tokens may contain very different information in these languages. This may make the partial reward not a good indicator of the final reward. It may better be noted that the proposed method might be exploiting the structure of English which is not universally true for all natural languages.
- Adding two hyperparameters to the inference algorithm is a minus (as noted in Limitations). They show in Tables 3 and 4 that the optimal values of the hyperparameters are relatively consistent when using different choices of reward models and the number of samples (N). The question is how consistent the choice of the tasks/prompts (which is noted in Prompt-dependent stopping) is. It would be nice if one could evaluate the robustness of the hyperparameters to the variety of tasks.
- Although the experiments are conducted on three language models and three reward models, they share relatively similar profiles. It would be beneficial to have larger and smaller language models and reward models.
- An empirical evaluation of the proposed method compared against MCTS would be nice to have.

**Questions:**

The speedup depends on the speed of the text generation and reward models. How fast are they in the experiments? It would be valuable to have the wall time of the text generation and reward computation separately in the Appendix.

**Limitations:**

I don't see any problems. If the method is likely to exploit the structure of English, then it may be noted in the limitations that the experiments are conducted only in English.

---

> ### Author Rebuttal · Authors · 2024-08-07
>
> # Response for the reviewer qQm8:
> We thank the reviewer for providing positive feedback and several good questions of our work. Below are our detailed responses.
>
> ---- "Experiments are conducted only on AlpacaFarm-Eval dataset. It would be ideal to have an evaluation of the proposed method in other datasets (e.g., TL;DR, hh-rlhf) as the effectiveness of the method may be affected by the structure of the task. For example, I speculate the the reward of the partial text has less correlation with the complete text on tasks like text summarization. It may also be influenced by the language. For example, Korean and Japanese have more flexible word order than English and word order does not determine grammatical function. Thus, the same length of the first few tokens may contain very different information in these languages. This may make the partial reward not a good indicator of the final reward. It may better be noted that the proposed method might be exploiting the structure of English which is not universally true for all natural languages." ----
>
> **Response:**
>
> In the attached PDF, we have several new experiments on the HH-RLHF dataset as well and we show that our algorithm performs consistently well across both tested datasets. We would have liked to also test on the TL;DR task, but could not due to limited time.
>
> Additionally, the reviewer makes a very good point regarding the specific semantics of the English language - in this sense, our method does take advantage of the structure of English and we will include this in the Limitations section.
>
> ---- "Adding two hyperparameters to the inference algorithm is a minus (as noted in Limitations). They show in Tables 3 and 4 that the optimal values of the hyperparameters are relatively consistent when using different choices of reward models and the number of samples (N). The question is how consistent the choice of the tasks/prompts (which is noted in Prompt-dependent stopping) is. It would be nice if one could evaluate the robustness of the hyperparameters to the variety of tasks." ---
>
> **Response:** In the attached PDF, we have tested the robustness of the hyperparameters by conducting multiple counterfactual analyses across several datasets, LMs, RMs, and tasks. We find that across all these combinations, an overwhelming majority of them use a decision token of either 128 or 256 and a rejection rate of 0.7 or 0.8. Then, a reasonable choice of hyperparameters in practice would be, say, rejecting 75% of the trajectories after generating 200 tokens. This would be a strong choice across a wide variety of real-world situations.
>
> ---- "Although the experiments are conducted on three language models and three reward models, they share relatively similar profiles. It would be beneficial to have larger and smaller language models and reward models." ---
>
> **Response:** In the attached PDF, we conduct several new experiments involving small and large LMs and RMs. We find that our method is effective across the spectrum of model sizes.
>
>
> ---- "An empirical evaluation of the proposed method compared against MCTS would be nice to have." ---
>
> **Response:**
> MCTS is definitely related as a decoding strategy. However, we could not complete the comparison by the end of the rebuttal period.
>
>
> ---- "The speedup depends on the speed of the text generation and reward models. How fast are they in the experiments? It would be valuable to have the wall time of the text generation and reward computation separately in the Appendix." ----
>
> **Response:** We can include detailed values in the appendix for all of the combinations, but it mostly depends on the maximum generation length specified by the LM. Rough estimates to produce and score a single batch of 20 on the new experiments are as follows on our machine:
>
> - GPT2-XL (max_length=1024): 25 seconds
> - GPT-J-6B (max_length=2048): 2-3 minutes
> - Mistral-7B (max_length=8000): 3-10 minutes
> - Llama-3-8B (max_length=8000): 1-5 minutes
>
> Reward model computations are generally 1-5 seconds, highlighting the intuition behind the effectiveness of our algorithm: because **reward computations are relatively cheap**, it is worth generating a few tokens (say, 200) and pruning most of the generations before continuing. Moreover, generating earlier tokens from decoder-only transformer architectures is much less expensive than generating later tokens due to the quadratic attention cost, compounding the strength of our method.

---

> > ### Comment · Reviewer_qQm8 · 2024-08-09
> >
> > Thank you very much for the clarification.
> >
> > I believe that the additional evaluation on the HH-RLHF dataset and on various language models further improved the reliability of the experimental results.
> >
> > > We can include detailed values in the appendix for all of the combinations, but it mostly depends on the maximum generation length specified by the LM. Rough estimates to produce and score a single batch of 20 on the new experiments are as follows on our machine:
> > > reward computations are relatively cheap
> >
> > I believe it is valuable to report the wall time (maybe in the appendix) even if it is a rough estimate. It will serve as evidence to say that reward computations are usually cheaper than the cost of generation.

---

> ### Comment · Area_Chair_5MpQ · 2024-08-12
> **Can you defend your review?**
>
> Dear Reviewer,
>
> This paper has been flagged as one with large discrepancies between scores. Could you please take a look at the other reviews and respond to the following question:
>
> Would you defend this paper getting accepting into NeurIPS this cycle?
>
> Thanks,
>
> AC

---

> > ### Comment · Reviewer_qQm8 · 2024-08-13
> >
> > I am quite positive that this paper brings an interesting contribution to the community.
> >
> > I am willing to discuss with Reviewer Qphu as their viewpoint is different from mine. However, I find it challenging to engage in a scientific discussion without supporting evidence or references for their claim.

---

### Official Review · Reviewer_Qphu · 2024-07-24

**Soundness:** 2
**Presentation:** 3
**Contribution:** 2
**Rating:** 5
**Confidence:** 4

**Summary:**

The paper proposes an early stopping method to accelerate the Best-of-N method. Experimental results demonstrate its effectiveness.

**Strengths:**

1. The method has a strong and clear motivation, coupled with easy implementation.
2. Experimental results demonstrate its effectiveness in accelerating best-of-n while preserving the quality of generated content.

**Weaknesses:**

1. Best-of-N is out-of-date in decoding-time alignment, which weakens the novelty of the paper.
2. The validity of the first point may be questionable when considering the utilization of best-of-n for response generation in model training. Nonetheless, the paper does not experiment with model training aided by SBoN.
3. It would be advantageous if the author could test SBoN on a broader range of tasks beyond just alignment.

**Questions:**

Please refer to the weakness part.

**Limitations:**

The authors acknowledge the challenge inherent in selecting hyperparameters while also offering solutions to address this issue.

---

> ### Author Rebuttal · Authors · 2024-08-07
>
> # Response for the reviewer Qphu:
>
> We thank the reviewer for providing valuable feedback and understanding the value of our work. We have read your comments carefully and below are our detailed responses.
>
> ---- "Best-of-N is out-of-date in decoding-time alignment, which weakens the novelty of the paper." ----
>
> **Response:** We find that Best-of-N(Rejection Sampling) is still a widely used technique that remains relevant even in recent times. For example, Song et al. [1] show that Best-of-N on relatively small models like Llama-3-8B-Instruct with values of N as small as 16-32 can outperform GPT-4-Turbo on several tasks. Additionally, Best-of-N was an important part of the deployment of OpenAI's WebGPT [2]. Also, rejection sampling is frequently used to generate high-quality data for alignment [3][4][5].
>
> Finally, very recent work has proposed aligning language models such that their distribution of generations is closer to the Best-of-N distribution [6][7].
>
> [1] Song, Yifan, et al. "The Good, The Bad, and The Greedy: Evaluation of LLMs Should Not Ignore Non-Determinism." arXiv preprint arXiv:2407.10457 (2024).
> [2] Nakano, Reiichiro, et al. "Webgpt: Browser-assisted question-answering with human feedback." arXiv preprint arXiv:2112.09332 (2021).
> [3] Khaki, Saeed, et al. "Rs-dpo: A hybrid rejection sampling and direct preference optimization method for alignment of large language models." arXiv preprint arXiv:2402.10038 (2024).
> [4] Liu, Tianqi, et al. "Statistical rejection sampling improves preference optimization." arXiv preprint arXiv:2309.06657 (2023).
> [5] Dubey, Abhimanyu, et al. "The Llama 3 Herd of Models." arXiv preprint arXiv:2407.21783 (2024).
> [6] Sessa, Pier Giuseppe, et al. "BOND: Aligning LLMs with Best-of-N Distillation." arXiv preprint arXiv:2407.14622 (2024).
> [7] Amini, Afra, et al. "Variational Best-of-N Alignment." arXiv preprint arXiv:2407.06057 (2024).
>
>
> ---- "The validity of the first point may be questionable when considering the utilization of best-of-n for response generation in model training. Nonetheless, the paper does not experiment with model training aided by SBoN." ----
>
> **Response:**
> As the reviewer suggests, best-of-n can be used both for inference-time alignment [2] or for later model fine-tuning [1].
> This latter setup is substantially more involved, and it introduces additional confounding factors (e.g., training hyper-parameters) which would have made the comparison harder.
> However, we added more metrics like the **win rate** computed by GPT4 [2] as a substitute (see attached PDF), to indicate that our algorithm produces high quality responses.
>
> [1] Dubey, Abhimanyu, et al. "The Llama 3 Herd of Models." arXiv preprint arXiv:2407.21783 (2024).
> [2] Dubois, Yann, et al. "Alpacafarm: A simulation framework for methods that learn from human feedback." Advances in Neural Information Processing Systems 36 (2024).
>
>
> ---- "It would be advantageous if the author could test SBoN on a broader range of tasks beyond just alignment." ----
>
> **Response:** In the attached PDF, we include preliminary experiments where we minimize the perplexity---which is equivalent to maximizing the probability of the generated text---with BoN and SBoN.
>
> We also include a few experiments in which we show that the GPT4 win rate remains relatively stable and large across rejection rates.
> Finally, we include several new experiments across multiple LMs and RMs of varying size as well as varying datasets to demonstrate the robustness of our method in different settings.
>
> Regarding the hyperparameter limitation, a significant majority of LM+RM combinations use a decision token of either 128 or 256 and a rejection rate of either 0.7 or 0.8 for Best-of-100. This suggests that rejecting, say, 75% of the trajectories at decision token 200 would be a simple and practical choice for hyperparameters in several real-world settings.

---

> > ### Comment · Reviewer_Qphu · 2024-08-14
> >
> > Thanks authors for your detailed reply and thanks reviewer qQm8 for pushing me into discussion.
> >
> > > Evidence for Best-of-N is out-of-date in **decoding-time alignment**
> >
> > Actually, you should provide me with evidence that they are widely used, not me. If they are not widely used, what evidence do you expect? You can see it from the authors' response. [1] is the only paper that uses Best-of-N, and it is a new paper. [2] is old, and [3] [4] [5] are about data generation. This means there is only 1 paper supporting their claims.
> >
> > > Addional Experiments
> >
> > Thanks for putting efforts into adding so many experiments and they are convincing. Therefore, I would like to change my score.

---

> ### Comment · Reviewer_qQm8 · 2024-08-07
> **Inquiry for the reference supporting the statement "Best-of-N is out-of-date"**
>
> > Best-of-N is out-of-date in decoding-time alignment
>
> I would like to ask Reviewer Qphu for the reference (or evidence) supporting this statement. It will be a more productive discussion if accompanied by evidence.

---

> ### Author Response · Authors · 2024-08-12
> **Reply to reviewer Qphu**
>
> Dear Reviewer Qphu,
>
> As the author-reviewer discussion period will end soon, we will appreciate it if you could check our response to your review comments. We have included all our new running experiments **in the attached PDF.** This way, if you have further questions and comments, we can still reply before the author-reviewer discussion period ends. If our response resolves your concerns, we kindly ask you to consider raising the rating of our work. Thank you very much for your time and efforts!
>
> The Authors

---

> ### Comment · Area_Chair_5MpQ · 2024-08-12
> **Reviewer, please respond to the authors**
>
> Hello reviewer,
>
> You wrote a very short review and gave a rating of "reject" which is much lower than the other two papers. Could you please engage with the authors and respond to their rebuttal? Do you stand by your review or would you like to change your score? Thanks for your assistance to the ACs with making a decision on this paper.
>
> -AC

---

### Author Rebuttal · Authors · 2024-08-07

# General Response to all reviewers:
We thank all reviewers for the detailed comments and valuable questions. We present additional experiments here---as requested by the reviewers---and also address the reviewers' individual questions separately.
These new experiments include:
- New datasets
- LLMs and reward models of different sizes
- Minimizing the perplexity as an experiment different from alignment
- GPT4 win rate experiments

Taken together, we believe that these should address several of the reviewers' concerns, and we hope they would consider improving their scores. If not, we are very happy to answer any further question that the reviewers may have.

## Experimental details

In the attached PDF we use our algorithm to minimize the **perplexity**, which is related to the probability of the generated text to showcase that the algorithm works in task different from alignment, as suggested by reviewer Qphu.

Moreover, we include new experimental results related to alignment across several datasets, as well as LLM and reward model sizes, as suggested by most reviewers.
- Datasets:
    - Alpaca Farm Eval
    - HH-RLHF
- Language Models (LMs):
    - GPT2-XL (1.5B)
    - GPT-J-6B
    - Mistral-7B
    - Llama-3-8B
- Reward Models (RMs):
    - reward-model-deberta-v3-large-v2 (~500M)
    - RM-Mistral-7B
    - FsfairX-Llama-3-RM (8B)
    - ArmoRM-Llama3-8B


Although we weren't able to bring all possible combinations to completion due to time constraints, these new experiments demonstrates that:
1. **Our method works** on a variety of model sizes. Speedups with negligible drop in score are apparent **across** the spectrum of **small and large LMs and RMs**.
2. **Our method is effective on several different tasks**. Notably, we test our method on new tasks that include minimizing generation perplexity. The effectiveness of our algorithm is relatively consistent across tasks.

---

### Decision · Program_Chairs · 2024-09-25

**Decision:**

Accept (poster)

**Comment:**

SUMMARY

Often people want to generate responses from a language model that have high scores according to some metric of interest.
The way this task is normally handled is by generating a bunch of possible outputs, scoring them according to the metric, and taking the best one.
The authors try to improve the efficiency of this strategy by halting the generation of unpromising utterances before they reach completion.

REASONS TO ACCEPT

The method the authors are trying to accelerate is a very common one that is used for a lot of tasks.
Therefore, the motivation for pursuing acceleration is very strong.
The method they come up with is simple and easy to implement, and the experimental results demonstrate its efficacy.

REASONS TO REJECT

- There are some concerns that best-of-N is out-of-date decoding strategy, which limits applicability of the method.
- One reviewer thought the experiments are insufficient, as they are exclusively conducted on the AlpacaFarm-Eval dataset, and it's unclear if the method will perform well on tasks with different structures languages. Also, all the LMs / reward models included in the experiments are pretty similar to each other.
- General comments about how organization and writing quality can be improved.

CONCLUSION

Even though the paper's impact might not be huge, it contains a nice, concentrated contribution which deserves to be shared with the NeurIPS community. However, the authors should try to resolve the organizational, missing citation, and writing suggestions mentioned by reviewers.